# Asymmetric Price Transmission: A Case of Wheat in India

**Ranjit Kumar Paul** [1,*] and **Tanmoy Karak** [2]

1  ICAR-Indian Agricultural Statistics Research Institute, New Delhi 110012, India
2  Upper Assam Advisory Center, Tea Research Association, Dikom 786101, India; t.karak@tocklai.net
*  Correspondence: ranjit.paul@icar.gov.in or ranjitstat@gmail.com

**Abstract:** In the present paper, horizontal and vertical integration was carried out on the wholesale and retail prices of wheat in the major markets of India. On confirming cointegration between the wholesale and retail prices of wheat in all needs, the vector error correction model (VECM) was applied to find the speed of adjustment in the corresponding price channel. The results revealed that price signals are transmitted across regions, indicating that price changes in one market are consistently related to price changes in markets and can influence the prices in other markets. In addition to studying cointegration, threshold autoregressive (TAR) and Momentum TAR (MTAR) models were applied to test for asymmetric cointegration. Hasen and Seo's test was used to test for the presence of threshold cointegration. It revealed a significant presence of asymmetric and nonlinear cointegration in many markets. Accordingly, a threshold VECM (TVECM) model with two regimes was applied. The results indicate that the retail price significantly responds to the deviations from the long-run equilibrium compared to the wholesale price.

**Keywords:** cointegration; error correction model; asymmetric price transmission; threshold cointegration; wheat

## 1. Introduction

India ranks second in the world after China regarding wheat production and consumption. India's share in global wheat production was recorded at 12.76 percent in 2016–2017. India has been self-sufficient in wheat production. India's share in global exports was around 0.40 percent in 2015–2016 (Directorate General of Commercial Intelligence and Statistics (DGCI&S). As per the 2016–2017 data collected from the Directorate of Economics and Statistics, Ministry of Agriculture and Farmers' Welfare, Government of India, it was revealed that the highest share of wheat comes from Uttar Pradesh (30% of total production), followed by Madhya Pradesh (18%), Punjab (17%), Haryana (12%), and Rajasthan (9%), which together share about 86% of the total national wheat production while Gujarat contributes the remaining 14% including Maharashtra, Uttarakhand, West Bengal, and other states of the country. The leading states in terms of area under wheat are Uttar Pradesh, Madhya Pradesh, Punjab, Haryana, and Rajasthan, accounting for more than 80 percent of wheat area in the country.

Market integration is a concept used to determine how markets of goods and services related to one another experience similar patterns in terms of an increase or decrease in the cost of commodities. The term can also be related to circumstances in which the costs of relatable goods and services sold in a defined geographical location also start to move in a similar pattern. Therefore, market integration is the phenomenon by which price interdependence takes place. Agricultural product markets established under market regulation programs play a vital role in providing marketplaces for farmers to sell their products. These markets also offer facilities and environments to the traders, processors, and other market functionaries for the smooth conduct of their trading activities. Agricultural commodities are characterized by seasonality, variability, perishability, etc. Poor

market integration is relatively uncompetitive. Scrappy and small farms with a low volume of marketable surplus makes the market difficult to perform.

The marketed surplus from production has also been rising, and it is estimated that about 60–70 percent of the production now comes to market. As a result, India's marketing system and efficiency are of significant concern and interest. Poor marketing efficiency can have severe consequences for both producers and consumers and affects government policy and the economy. Significant concerns have been raised regarding working the market mechanisms and market-related policies for wheat, a staple food crop. The market price of grain necessarily influences the demand for wheat. The demand-side market of wheat products needs to be explored to understand the impact on the production and growth of the crop. It is necessary to look at the price movement of wheat in different markets due to mismatch between demand–supply and large-scale imports.

Market integration is generally composed of two types: horizontal market integration and vertical market integration. Horizontal market integration indicates that the price of a commodity in one market responds to a change in the price of the same product in other markets. Vertical market integration represents the integration of the cost of the same product at different levels of the value chain (farm price, wholesale price, and retail price).

Usually, market reforms are required in order to achieve efficient agricultural markets and production systems. The producers and consumers in the marketing chain can realize the potential gains from the common market when the markets are integrated. In this regard, the prices of wheat play a vital role. When the markets are integrated, the price signals are transmitted from one call to another and influence other market prices. The wholesale market prices (producer's level) and retail market (consumer's level) prices are essential components of the marketing process. Studying the market integration of wheat has become necessary due to the interdependence of wholesale and retail markets. Price transmission can be of two types: symmetric and asymmetric. In contrast to symmetric price transmission, asymmetric price transmission (APT) is said to exist when the adjustment of prices is not homogeneous concerning external or internal characteristics of the system. Significant causes of asymmetric price transmission are non-competitive markets and adjustment costs [1]. Perez-Mesa et al. [2] investigated retail price rigidity in perishable food products and reported that price asymmetry was not avoided despite the relaxation of bargaining power.

## 2. Background

In the literature, several empirical studies have been carried out using cointegration techniques that concern the market integration of agricultural commodities in India and other countries. Asche et al. [3] carried out vertical and horizontal price linkages for salmon and found a high degree of price transmission in both supply chains and integrated markets in salmon. The price convergence across various regions in India was investigated by [4] and their results indicated a significant presence of cross-sectional dependence in prices in India. Jha et al. [5] stated that local food scarcity would persist if agricultural markets are not integrated. Balaguer and Ripollés [6] studied integration among transport fuel retail markets in Spain, and the future trading of soybean in India using a cointegration approach was studied in [7]. The non-linear error correction model was applied to assess price linkages and patterns of transmission among producer and consumer markets for apple in Slovenia [8]. Paul et al. [9] studied the volatility and associations between domestic and export prices of onion in India. The market integration of coffee prices in primary markets of India was studied by [10]. Gricar and Bojnec [11] investigated the causalities between Slovenian tourism and macroeconomic aggregate, and they reported that input costs were significant drivers for the rise in Slovenian hospitality industry prices. Ricci et al. [12] studied the effects of price instability in agriculture on vertical price transmission in Italy. The market integration using monthly price data for 21 agricultural goods and 60 markets in India was carried out by [13]. The authors reported no robust evidence that price integration has increased in recent years. Gricar et al. [14] applied the vector error

correction model (VECM) and found that hospitality prices in Montenegro were domestic driven and in Slovenia Eurozone driven. However, little work has been carried out on empirically evaluating wheat market integration in India. Empirical evaluation of intrastate and inter-state spatial integration of wheat markets in India was carried out in [15], where strong evidence favoring the spatial integration of regional wheat markets was reported. Gandhi and Zhou [16] indicated that in India, wheat production is concentrated, and growth is driven predominantly by yield increases, and to some extent, by a shift in the area from other crops. Mukim et al. [17] established that the wheat markets are integrated across states of India in the long-run, but not in the short-run. All of these studies concentrated on the spatial (horizontal) market integration of wheat; there is hardly any work related to India's vertical integration of wheat prices.

However, unlike the above studies, the prices of many commodities including wheat are characterized by asymmetric adjustment [18,19]. Meyer and Von Cramon-Taubadel [1] studied the different types and causes of asymmetric price transmission and described the econometric techniques used to quantify it. Enders and Granger [20] investigated the asymmetric movements toward long-run equilibrium. In this situation, it is essential to use the threshold cointegration approach, which allows for the asymmetric adjustment introduced by [21]. The authors reported that if price adjustment is asymmetric, then the standard cointegration tests and extensions are not correctly specified. In the literature, estimation methods of threshold cointegration have been extensively studied [22,23]. The structure of interest rates by the threshold cointegration model was studied in [24]. The price–transmission dynamics in the Iranian egg market was studied by [25]. Asymmetric panel vector error correction model (VECM) was applied to study the price transmission mechanism along the European food supply chain [26]. Price asymmetry in Bangladesh rice markets was studied by [27]. The nonlinear autoregressive distributed lag (NARDL) model was utilized to study the rice trade in Southeast Asia [28]. Threshold cointegration and asymmetric error-correction approaches were applied to investigate the nexus between India's current and capital accounts [29]. However, the application of threshold cointegration in agriculture is scarce. The present study deals with market integration in wheat prices both horizontally and vertically. Moreover, there is no study of asymmetric cointegration in wheat prices in Indian markets. Therefore, the present study attempts to examine the movement of wheat prices in different markets across the states of India and the transmission of price signals and information across these markets. For this purpose, threshold autoregressive (TAR) and momentum-TAR (MTAR) approaches have been applied [30]. A good description of asymmetric integration may be found in [31].

## 3. Data and Methodology

The study was undertaken covering seventeen major markets: Delhi, Jammu, Amritsar, Ludhiana, Lucknow, Dehradun, Raipur, Ahmedabad, Bhopal, Mumbai, Jaipur Patna, Bhubaneswar, Bengaluru, Thiruvananthapuram, Chennai, and Hyderabad, along with the all India maximum, minimum, and modal price of wheat. Daily data on retail and wholesale prices of wheat of the above markets from January 2010 to May 2018 were collected from the Department of Consumer Affairs, Government of India [32]. The Jarque–Bera test statistics were used to check the normality of the series. Wholesale price is the price at which when the retailers buy products in large volumes. Retail prices are what retailers set as the final selling price for consumers. The daily data were converted into weekly data. The missing observations were imputed by using the mean value (i.e., if the 3rd week of January 2015 is missing, it is replaced by the mean of the 3rd week price of January of preceding years). The econometric methods used in the present investigation are described below in brief. These techniques allow one to quantify the degree of interconnectedness between the markets. In order to check for the presence of unit root, the tests, namely the augmented Dickey–Fuller (ADF) test [33], Phillips–Perron unit root test [34], and KPSS test [35] were used.

### 3.1. Johansen's Approach to Cointegration

The multivariate cointegration approach [36] examines the cointegration among price series. Let $y_t$ be n × 1 set of I(1) variables (if the series is integrated to order d, it is denoted as I(d)). Usually, any linear combination $a'y_t$ will be I(1) for a ≠ 0. However, if there exists an n × 1 vector $\alpha_i$ such that $\alpha_i'y_t$ is (0), $\alpha_i \neq 0$, then it is said that the components in $y_t$ are cointegrated of order one, denoted CI(1) with the cointegrating vector as $\alpha_i$. It can be noted that if $\alpha_i$ is a cointegrating vector, then so is the $k\alpha_i$ for any k ≠ 0, since $k\alpha_i' y_t \sim I(0)$.

For the total n series, there can be maximum r different cointegrating vectors, where $0 \leq r = k - 1 < n$.

### 3.2. Error Correction Models (ECM)

The ECM can be written as per Equation (1) in vector and matrix notation.

$$\Delta y_t = \alpha\beta'y_{t-1} + \Gamma_1\Delta y_{t-1} + u_t \tag{1}$$

where $\alpha = [\alpha 1, \alpha 2]'$, $\beta' = [1, -\beta 1]$, and $\Gamma_1 = \begin{bmatrix} \gamma_{11} & \gamma_{12} \\ \gamma_{21} & \gamma_{22} \end{bmatrix}$.

Equation (1) can be reformulated into a vector error correction model (VECM):

$$\Delta y_t = \Pi y_{t-1} + \sum_{j=1}^{k-1} \Gamma_j \Delta y_{t-j} + u_t, \ t = k+1, \ldots, T \tag{2}$$

where $\Gamma_i = -(A_{i+1} + \ldots + A_k)$, I = 1, ..., k − 1, and $\Pi = -(I - A_1 - \ldots - A_k)$. The estimates $\hat{\Gamma}_i$ and $\hat{\Pi}$, respectively explain the short and long-run adjustments to changes in $y_t$.

Here, $\Pi = \alpha\beta'$, where $\alpha$ represents the rate of adjustments and $\beta$ is a matrix of long-run coefficients. The optimum lag length is selected based on the minimum value of the information criterion (e.g., Akaike information criterion (AIC) and Baysian information criterion (BIC)).

### 3.3. Asymmetric Cointegration

The standard cointegration tests and their extensions are mis-specified if the adjustment is asymmetric [21]. Furthermore, to capture asymmetry in a cointegrating relationship, the authors extended the threshold autoregressive (TAR) and momentum threshold autoregressive (M-TAR) models of [20] to a multivariate context. The aspects of 'deep movements' can be captured by the TAR model while 'steep movements' in a price is captured by M-TAR ([20]). Hassanzoy et al. [19] reported that the M-TAR model is superior to that of TAR and the Engle and Granger tests. In the present investigation, in order to take care of the large changes in the price series, the M-TAR model was applied. The consistent M-TAR model is defined by Equations (3)–(5). Here, the speed of adjustment toward equilibrium depends on the direction of change in $\hat{\varepsilon}_{t-1}$, that is, $\Delta\hat{\varepsilon}_{t-1}$. Therefore, the speed of adjustment is $\rho_1\hat{\varepsilon}_{t-1}$, if deviations from the long-run equilibrium are positive, and $\rho_2\hat{\varepsilon}_{t-1}$ otherwise.

$$\Delta\hat{\varepsilon}_t = I_t\rho_1\hat{\varepsilon}_{t-1} + (1 - I_t)\rho_2\hat{\varepsilon}_{t-1} + \sum_{i=1}^{p-1} \beta_i\Delta\hat{\varepsilon}_{t-i} + \omega_t \tag{3}$$

$$I_t = \begin{cases} 1 & if \ \Delta\hat{\varepsilon}_{t-1} \geq a_0 \\ 0 & if \ \Delta\hat{\varepsilon}_{t-1} < a_0 \end{cases} \tag{4}$$

where $a_0$ is a threshold value; $\rho_1$ and $\rho_2$ are adjustment coefficients; $\beta_i$ indicates the coefficient(s) of lagged changes; and $\omega_t$ is the identically independently distrusted (i.i.d.) stochastic term. The necessary and sufficient conditions for stationarity of $\hat{\varepsilon}_t$ are $\rho_1 < 0$, $\rho_2 < 0$, and $(1 + \rho_1)(1 + \rho_2) < 1$ for any values of $a_0$ [18,19]. Tong [37] showed that the least square estimates of $\rho_1$ and $\rho_2$ had an asymptotic multivariate normal distribution under the condition that $\hat{\varepsilon}_t$ is stationary. The M-TAR model was applied in this paper to examine the long-run relationship among the pairs of wholesale and retail prices of wheat assuming asymmetric adjustment. For threshold cointegration with M-TAR adjustment, the five-step

procedure reported in [19] was followed. First, a long-run relationship between the pairs of markets was estimated as follows:

$$y_{1,t} = \gamma_0 + \gamma_1 y_{2,t} + \varepsilon_t \tag{5}$$

where $y_{1,t}$ and $y_{2,t}$ are the logarithm of wholesale and retail prices of wheat at time t; $\gamma_0$ is a constant term; $\gamma_1$ is th elasticity of price transmission; and $\varepsilon_t$ is the error term that may be serially correlated. In the second step, following [38], consistent estimates of threshold values for M-TAR models were obtained. Equations (3) and (4) were estimated for each of the possible threshold values. Finally, the threshold ($a_0$) was estimated by minimizing the sum of squared residuals from the fitted model. In the third step, the null hypothesis of no cointegration, that is, $\rho_1 = \rho_2 = 0$, was tested for each of the M-TAR models. Fourth, the null hypothesis of no asymmetric adjustment, that is, $\rho_1 = \rho_2$, was tested for each of the M-TAR models using the standard F-test under the condition that the null hypothesis of no cointegration is rejected. In the last step, the Ljung–Box Q-statistic was applied to test for the white noise process of the estimated residuals from the M-TAR models. Once the presence of cointegration was established among the markets, the dynamics of price transmission among them were analyzed using asymmetric vector error correction models (AVECMs) with threshold (M-TAR) adjustment. On application of the TAR model, it revealed that asymmetric cointegration was present in the wholesale and retail prices of wheat in the markets, namely Ahmedabad, Bengaluru, Bhubaneswar, Hyderabad, Patna and the all India minimum price. On the other hand, using the MTAR model, it was seen that asymmetric cointegration was present in most markets.

## 4. Results and Discussion

Summary statistics of the price data of different markets were computed and are reported in Table 1. It can be seen that both the wholesale and retail prices were more or less consistent in the markets as depicted by the coefficient of variation (CV) value. In the market integration study, the first step is to check for evidence of the non-stationarity of data. Test for stationarity was performed by using the ADF test, PP test, and the KPSS test. The results of all three tests, as reported in Table 2, revealed that all variables were non-stationary. In order to achieve stationarity, the series were differenced to the first order and all series became stationary after the first differentiation. As all the series were found to be integrated of the same order, the dataset was suitable for cointegration.

**Table 1.** Descriptive statistics for the wholesale and retail prices (INR per quintal) of the individual markets.

| Markets | Wholesale Price | | | | | | Retail Price | | | | | |
|---|---|---|---|---|---|---|---|---|---|---|---|---|
| | Mean | Median | Max | Min | SD | CV(%) | Mean | Median | Max | Min | SD | CV(%) |
| Ahmedabad | 1647.75 | 1650.00 | 2000.00 | 1100.00 | 302.89 | 18.38 | 1851.49 | 1900.00 | 2300.00 | 1200 | 318.11 | 17.18 |
| Amritsar | 1491.28 | 1500.00 | 1900.00 | 1050.00 | 229.44 | 15.39 | 1688.93 | 1800.00 | 2200.00 | 1100 | 297.61 | 17.62 |
| Bengaluru | 2320.10 | 2500.00 | 2800.00 | 1671.43 | 418.44 | 18.04 | 2582.08 | 2700.00 | 3400.00 | 1800 | 530.42 | 20.54 |
| Bhopal | 1457.20 | 1500.00 | 1700.00 | 1050.00 | 170.97 | 11.73 | 1638.73 | 1700.00 | 2000.00 | 1100 | 234.91 | 14.33 |
| Bhubaneswar | 1512.65 | 1580.00 | 1580.00 | 1210.00 | 107.36 | 7.10 | 1803.18 | 1800.00 | 2014.29 | 1400 | 222.5 | 12.34 |
| Chennai | 2298.40 | 2342.86 | 2814.29 | 1800.00 | 204.42 | 8.89 | 2794.45 | 3000.00 | 3500.00 | 2000 | 439.36 | 15.72 |
| Dehradun | 1511.74 | 1512.86 | 1880.00 | 1120.00 | 218.85 | 14.48 | 1709.16 | 1600.00 | 2200.00 | 1200 | 310.24 | 18.15 |
| Delhi | 1590.37 | 1671.43 | 2227.86 | 1138.57 | 244.90 | 15.40 | 1800.62 | 1900.00 | 2414.29 | 1300 | 234.88 | 13.04 |
| Hyderabad | 2359.37 | 2400.00 | 2700.00 | 1490.71 | 356.55 | 15.11 | 2581.75 | 2700.00 | 2900.00 | 1700 | 353.88 | 13.71 |
| Jaipur | 1570.16 | 1600.00 | 2250.00 | 1150.00 | 233.97 | 14.90 | 1709.90 | 1700.00 | 2600.00 | 1300 | 256.55 | 15.00 |
| Jammu | 1562.63 | 1617.14 | 2560.00 | 1000.00 | 253.82 | 16.24 | 1698.18 | 1700.00 | 2200.00 | 1100 | 259.59 | 15.29 |
| Lucknow | 1445.64 | 1450.00 | 1900.00 | 1035.71 | 215.18 | 14.88 | 1576.59 | 1600.00 | 2000.00 | 1100 | 213.26 | 13.53 |

**Table 1.** *Cont.*

| Markets | Wholesale Price | | | | | | Retail Price | | | | | |
|---|---|---|---|---|---|---|---|---|---|---|---|---|
| | Mean | Median | Max | Min | SD | CV(%) | Mean | Median | Max | Min | SD | CV(%) |
| Ludhiana | 1489.15 | 1350.00 | 1900.00 | 1114.29 | 242.65 | 16.29 | 1601.62 | 1592.86 | 2000.00 | 1200 | 243.15 | 15.18 |
| Mumbai | 2102.12 | 2200.00 | 2628.57 | 1453.57 | 313.44 | 14.91 | 2614.76 | 2700.00 | 3514.29 | 1700 | 447.64 | 17.12 |
| Patna | 1518.12 | 1600.00 | 2200.00 | 1100.00 | 238.40 | 15.70 | 1730.36 | 1800.00 | 2400.00 | 1200 | 292.66 | 16.91 |
| Thiruvananthapuram | 2331.98 | 2400.00 | 3200.00 | 1500.00 | 405.95 | 17.41 | 2557.70 | 2600.00 | 3500.00 | 1700 | 413.19 | 16.15 |
| Maximum Price | 2833.35 | 2850.00 | 4021.43 | 1833.00 | 546.88 | 19.30 | 3109.81 | 3100.00 | 4428.57 | 2100 | 632.25 | 20.33 |
| Minimum Price | 1276.89 | 1337.14 | 1471.43 | 100.00 | 169.09 | 13.24 | 1401.84 | 1500.00 | 1671.43 | 1000 | 177.20 | 12.64 |
| Modal Price | 1717.69 | 1667.86 | 2714.29 | 1100.00 | 395.82 | 23.04 | 1712.87 | 1797.64 | 2478.57 | 1200 | 267.59 | 15.62 |

**Table 2.** Stationarity test results.

| Markets | Original Series | | | Differenced Series | | |
|---|---|---|---|---|---|---|
| | Wholesale Price | | | | | |
| | ADF Test Statistic | PP Test Statistic | KPSS Test Statistic | ADF Test Statistic | PP Test Statistic | KPSS Test Statistic |
| Ahmedabad | −1.43 | −1.44 | 114.11 * | −15.56 * | −15.74 * | 0.89 |
| Amritsar | −0.46 | −0.49 | 136.34 * | −14.92 * | −14.77 * | 1.46 |
| Bengaluru | −0.70 | −0.55 | 116.30 * | −15.37 * | −16.64 * | 0.99 |
| Bhopal | −0.71 | −0.82 | 178.78 * | −15.13 * | −14.31 * | 0.88 |
| Bhubaneswar | −1.89 | −1.77 | 295.55 * | −14.00 * | −13.90 * | 0.29 |
| Chennai | −2.48 | −2.78 | 235.84 * | −17.01 * | −14.80 * | 0.46 |
| Dehradun | −0.90 | −0.69 | 144.90 * | −13.66 * | −13.28 * | 1.16 |
| Delhi | −1.56 | −1.46 | 136.22 * | −15.00 * | −15.04 * | 0.75 |
| Hyderabad | −2.13 | −2.10 | 138.80 * | −18.50 * | −18.50 * | 1.22 |
| Jaipur | −1.71 | −1.66 | 140.77 * | −14.26 * | −14.27 * | 0.89 |
| Jammu | −1.80 | −2.45 | 129.14 * | −15.21 * | −21.69 * | 0.33 |
| Lucknow | −1.75 | −1.50 | 140.92 * | −13.14 * | −13.86 * | 1.02 |
| Ludhiana | −0.15 | −0.19 | 128.73 * | −15.79 * | −21.58 * | 1.31 |
| Maximum Price | −0.82 | −1.58 | 108.68 * | −18.52 * | −77.21 * | 0.71 |
| Minimum Price | −2.89 | −3.14 | 158.40 * | −20.26 * | −22.65 * | -0.01 |
| Modal Price | −2.64 | −3.95 | 91.03 * | −12.85 * | −41.55 * | 0.03 |
| Mumbai | −1.37 | −1.02 | 140.68 * | −14.51 * | −15.25 * | 0.89 |
| Patna | −2.01 | −2.06 | 133.58 * | −23.90 * | −24.12 * | 0.33 |
| Thiruvananthapuram | −1.97 | −2.10 | 120.50 * | −17.09 * | −16.53 * | 0.31 |
| | Retail Price | | | | | |
| Ahmedabad | −1.32 | −1.24 | 122.09 * | −14.47 * | −17.00 * | 0.31 |

**Table 2.** *Cont.*

| Markets | Original Series | | | Differenced Series | | |
|---|---|---|---|---|---|---|
| | **Wholesale Price** | | | | | |
| | **ADF Test Statistic** | **PP Test Statistic** | **KPSS Test Statistic** | **ADF Test Statistic** | **PP Test Statistic** | **KPSS Test Statistic** |
| Amritsar | −1.43 | −1.23 | 119.04 * | −14.73 * | −14.08 * | 0.98 |
| Bengaluru | −0.32 | −0.26 | 102.11 * | −15.56 * | −15.52 * | 1.53 |
| Bhopal | −0.94 | −1.02 | 146.33 * | −10.61 * | −14.20 * | 0.91 |
| Bhubaneswar | −0.84 | −0.83 | 170.00 * | −13.44 * | −13.33 * | 0.89 |
| Chennai | −1.59 | −1.58 | 133.41 * | −15.52 * | −14.54 * | 1.08 |
| Dehradun | −0.69 | −0.79 | 115.56 * | −12.12 * | −13.87 * | 1.02 |
| Delhi | −2.02 | −1.43 | 160.81 * | −13.96 * | −13.72 * | 0.70 |
| Hyderabad | −2.55 | −2.49 | 153.03 * | −13.80 * | −13.09 * | 1.44 |
| Jaipur | −2.07 | −2.27 | 139.81 * | −15.30 * | −18.30 * | 0.51 |
| Jammu | −1.38 | −1.27 | 137.22 * | −9.34 * | −15.40 * | 0.70 |
| Lucknow | −1.45 | −1.45 | 155.07 * | −16.67 * | −15.30 * | 0.79 |
| Ludhiana | −0.92 | −0.66 | 138.17 * | −16.03 * | −18.52 * | 0.95 |
| Maximum Price | −0.30 | −0.09 | 103.17 * | −19.55 * | −31.50 * | 0.99 |
| Minimum Price | −1.83 | −1.69 | 165.94 * | −20.83 * | −20.98 * | 0.22 |
| Modal Price | −2.27 | −3.24 | 134.27 * | −21.57 * | −62.84 * | 0.20 |
| Mumbai | −1.37 | −1.23 | 122.53 * | −12.47 * | −17.59 * | 0.85 |
| Patna | −1.55 | −1.42 | 124.02 * | −10.28 * | −16.48 * | 1.24 |
| Thiruvananthapuram | −1.80 | −1.85 | 129.85 * | −16.67 * | −18.33 * | 0.13 |

* Denotes test is significant at 5% level of significance.

*Cointegration in Price Series*

Johansen's cointegration test has been applied to investigate cointegration among different markets with respect to wholesale and retail prices. For horizontal integration, 11 markets were selected based on the production and consumption of wheat in different states of India. The 11 selected markets were: Ahmedabad, Amritsar, Bhopal, Dehradun, Delhi, Jaipur, Jammu, Lucknow, Ludhiana, Mumbai, and Patna. It revealed that markets were perfectly cointegrated with respect to wholesale as well as retail price. Both the maximum eigenvalue statistic and trace statistics were used to test the cointegration and the results are reported in Table 3. In retail price, according to the trace statistics, the no. of cointegrating equations was six, whereas the eigen value statistics indicate that the no. of cointegrating equations was two. Similarly, for wholesale price, no. of cointegrating equations was found to be four and three, respectively, based on the trace and eigenvalue statistics. In addition to the horizontal cointegration, the vertical cointegration between the wholesale and retail prices of wheat in an individual market was also investigated. The results of Johansen's cointegration test are presented in Table 4 using the trace and eigen statistics. It was observed that the wholesale and retail prices were integrated in all markets.

**Table 3.** Cointegration among the retail and wholesale prices of wheat.

| No. of Cointegrating Equations | Retail Price | | | |
|---|---|---|---|---|
| | **Test Statistics (Trace)** | **5% Critical Value** | **Test Statistics (Eigen)** | **5% Critical Value** |
| None | 392.86 | 277.39 | 95.04 | 68.27 |
| At most 1 | 297.82 | 232.49 | 75.47 | 62.42 |
| At most 2 | 222.35 | 192.84 | 53.70 | 57.00 |
| At most 3 | 168.65 | 157.11 | 40.3 | 51.07 |
| At most 4 | 128.35 | 124.25 | 37.76 | 44.91 |
| At most 5 | 90.59 | 90.39 | 33.95 | 39.43 |
| At most 6 | 56.64 | 70.60 | 23.41 | 33.32 |
| At most 7 | 33.23 | 48.28 | 14.69 | 27.14 |
| At most 8 | 18.54 | 31.52 | 11.45 | 21.07 |
| At most 9 | 7.09 | 17.95 | 6.91 | 14.90 |
| At most 10 | 0.18 | 8.18 | 0.18 | 8.18 |
| | Wholesale Price | | | |
| None | 438.07 | 277.39 | 126.4 | 68.27 |
| At most 1 | 311.67 | 232.49 | 79.39 | 62.42 |
| At most 2 | 232.28 | 192.84 | 73.67 | 57.00 |
| At most 3 | 158.61 | 157.11 | 50.88 | 51.07 |
| At most 4 | 107.73 | 124.25 | 34.12 | 44.91 |
| At most 5 | 73.61 | 90.39 | 25.90 | 39.43 |
| At most 6 | 47.71 | 70.60 | 17.71 | 33.32 |
| At most 7 | 30.00 | 48.28 | 14.84 | 27.14 |
| At most 8 | 15.16 | 31.52 | 8.71 | 21.07 |
| At most 9 | 6.45 | 17.95 | 6.34 | 14.90 |
| At most 10 | 0.11 | 8.18 | 0.11 | 8.18 |

**Table 4.** Market-wise cointegration between the wholesale and retail prices of wheat.

| No. of Cointegrating Equations | Eigen Value | Test Statistics (Eigen) | 5% Critical Value | Test Statistics (Trace) | 5% Critical Value |
|---|---|---|---|---|---|
| | | Delhi | | | |
| None | 0.0054 | 22.43 | 14.90 | 24.85 | 17.95 |
| At most 1 | 0.0499 | 2.41 | 8.18 | 2.41 | 8.18 |
| | | Ahmedabad | | | |
| None | 0.0061 | 43.77 | 14.90 | 46.48 | 17.95 |
| At most 1 | 0.0951 | 2.71 | 8.18 | 2.71 | 8.18 |
| | | Amritsar | | | |
| None | 0.0023 | 9.54 | 14.90 | 10.58 | 17.95 |
| At most 1 | 0.0215 | 1.04 | 8.18 | 1.04 | 8.18 |
| | | Bengaluru | | | |
| None | 0.0016 | 8.75 | 14.90 | 9.47 | 17.95 |
| At most 1 | 0.0197 | 0.72 | 8.18 | 0.72 | 8.18 |

**Table 4.** *Cont.*

| No. of Cointegrating Equations | Eigen Value | Test Statistics (Eigen) | 5% Critical Value | Test Statistics (Trace) | 5% Critical Value |
|---|---|---|---|---|---|
| | | Bhopal | | | |
| None | 0.0030 | 42.07 | 14.90 | 43.40 | 17.95 |
| At most 1 | 0.0915 | 1.33 | 8.18 | 1.33 | 8.18 |
| | | Bhubaneswar | | | |
| None | 0.0025 | 14.63 | 14.90 | 15.73 | 17.95 |
| At most 1 | 0.0328 | 1.10 | 8.18 | 1.10 | 8.18 |
| | | Chennai | | | |
| None | 0.0070 | 25.69 | 14.90 | 28.77 | 17.95 |
| At most 1 | 0.0569 | 3.09 | 8.18 | 3.09 | 8.18 |
| | | Dehradun | | | |
| None | 0.0026 | 13.67 | 14.90 | 14.84 | 17.95 |
| At most 1 | 0.0307 | 1.16 | 8.18 | 1.16 | 8.18 |
| | | Hyderabad | | | |
| None | 0.0089 | 37.59 | 14.90 | 41.53 | 17.95 |
| At most 1 | 0.0822 | 4.00 | 8.18 | 4.00 | 8.18 |
| | | Jaipur | | | |
| None | 0.0055 | 58.65 | 14.90 | 61.09 | 17.95 |
| At most 1 | 0.1253 | 2.44 | 8.18 | 2.44 | 8.18 |
| | | Jammu | | | |
| None | 0.0064 | 134.00 | 14.90 | 136.80 | 17.95 |
| At most 1 | 0.2634 | 2.84 | 8.18 | 2.84 | 8.18 |
| | | Lucknow | | | |
| None | 0.0068 | 52.58 | 14.90 | 55.61 | 17.95 |
| At most 1 | 0.1131 | 3.03 | 8.18 | 3.03 | 8.18 |
| | | Ludhiana | | | |
| None | 0.0015 | 31.19 | 14.90 | 31.88 | 17.95 |
| At most 1 | 0.0687 | 0.69 | 8.18 | 0.69 | 8.18 |
| | | Mumbai | | | |
| None | 0.0049 | 33.22 | 14.90 | 35.40 | 17.95 |
| At most 1 | 0.0730 | 2.18 | 8.18 | 2.18 | 8.18 |
| | | Patna | | | |
| None | 0.0061 | 21.00 | 14.90 | 23.67 | 17.95 |
| At most 1 | 0.0467 | 2.69 | 8.18 | 2.69 | 8.18 |
| | | Thiruvananthapuram | | | |
| None | 0.0101 | 52.46 | 14.90 | 56.95 | 17.95 |
| At most 1 | 0.1128 | 4.49 | 8.18 | 4.49 | 8.18 |

Source: Authors' estimation.

Before investigating the presence of asymmetric cointegration, the presence of non-linearity was tested using the BDS test [39]. The results of the BDS test is reported in Table 5. The results indicate that all of the series were nonlinear in nature. After assuring cointegration among the wholesale and retail prices of wheat, test of the presence of asymmetric cointegration was investigated by means of the MTAR model, as described in Section 3.3. The results of the asymmetric cointegration tests are presented in Table 6. In Table 6, Phi determines whether retail and wholesale prices are cointegrated or not and APT (asymmetric price transmission) checks whether price transmission between individual

markets of retail and wholesale price is of a symmetric or asymmetric nature. The MTAR model revealed that Delhi, Ahmedabad, Bhopal, Chennai, Hyderabad, Jaipur, Jammu, Lucknow, Ludhiana, Maximum, Minimum, and Modal series had the property of APT.

**Table 5.** BDS test to test the nonlinearity in each of the price series.

| Markets | Dimension | Epsilon (1) | Epsilon (2) | Epsilon (3) | Epsilon (4) |
|---|---|---|---|---|---|
| Ahmedabad_Retail | 2 | 600.67 | 173.78 | 117.59 | 70.00 |
| | 3 | 1129.23 | 215.99 | 133.03 | 70.30 |
| Ahmedabad_Wholesale | 2 | 2173.76 | 319.87 | 108.27 | 72.50 |
| | 3 | 3950.85 | 431.49 | 119.93 | 72.39 |
| Amritsar_Retail | 2 | 93.81 | 117.40 | 91.05 | 69.29 |
| | 3 | 149.09 | 160.84 | 102.21 | 69.63 |
| Amritsar_Wholesale | 2 | 180.06 | 136.23 | 88.66 | 73.80 |
| | 3 | 329.51 | 181.74 | 96.50 | 71.81 |
| Bengaluru_Retail | 2 | 292.59 | 323.93 | 131.92 | 59.34 |
| | 3 | 522.18 | 439.46 | 147.52 | 57.72 |
| Bengaluru_Wholesale | 2 | 244.40 | 1255.52 | 181.84 | 72.96 |
| | 3 | 365.17 | 1663.65 | 217.31 | 73.67 |
| Bhopal_Retail | 2 | 137.42 | 165.38 | 101.25 | 62.23 |
| | 3 | 222.07 | 220.16 | 111.85 | 61.05 |
| Bhopal_Wholesale | 2 | 209.93 | 180.14 | 101.56 | 64.07 |
| | 3 | 421.36 | 247.94 | 111.55 | 62.80 |
| Bhubaneshwar_Retail | 2 | 166.81 | 84.49 | 74.55 | 61.24 |
| | 3 | 277.31 | 100.98 | 81.61 | 62.74 |
| Bhubaneshwar_Wholesale | 2 | 36.89 | 51.99 | 45.93 | 35.62 |
| | 3 | 47.80 | 63.11 | 50.41 | 35.26 |
| Chennai_Retail | 2 | 181.38 | 187.00 | 103.10 | 73.61 |
| | 3 | 295.91 | 241.48 | 116.06 | 74.61 |
| Chennai_Wholesale | 2 | 100.65 | 85.42 | 63.95 | 53.87 |
| | 3 | 156.00 | 103.09 | 67.05 | 51.98 |
| Dehradun_Retail | 2 | 669.71 | 166.61 | 107.52 | 73.71 |
| | 3 | 1230.74 | 210.35 | 121.92 | 72.53 |
| Dehradun_Wholesale | 2 | 539.88 | 199.04 | 90.49 | 75.48 |
| | 3 | 1033.83 | 276.57 | 96.11 | 73.98 |
| Delhi_Retail | 2 | 88.11 | 91.31 | 72.94 | 61.73 |
| | 3 | 135.25 | 115.87 | 80.75 | 61.20 |
| Delhi_Wholesale | 2 | 113.22 | 92.92 | 83.71 | 70.42 |
| | 3 | 177.46 | 118.92 | 95.89 | 70.09 |
| Hyderabad_Retail | 2 | 72.33 | 65.15 | 57.72 | 50.04 |
| | 3 | 118.83 | 80.06 | 62.26 | 49.80 |
| Hyderabad_Wholesale | 2 | 103.98 | 69.05 | 64.85 | 52.94 |
| | 3 | 185.87 | 83.48 | 71.01 | 53.02 |

**Table 5.** *Cont.*

| Markets | Dimension | Epsilon (1) | Epsilon (2) | Epsilon (3) | Epsilon (4) |
|---|---|---|---|---|---|
| Jaipur_Retail | 2 | 158.78 | 97.03 | 65.89 | 52.97 |
| | 3 | 277.57 | 125.27 | 71.73 | 50.48 |
| Jaipur_Wholesale | 2 | 183.13 | 102.85 | 72.85 | 64.64 |
| | 3 | 315.12 | 133.01 | 79.83 | 63.58 |
| Jammu_Retail | 2 | 111.30 | 129.08 | 68.34 | 66.87 |
| | 3 | 194.38 | 173.25 | 76.45 | 63.79 |
| Jammu_Wholesale | 2 | 109.77 | 99.42 | 68.60 | 58.83 |
| | 3 | 191.54 | 129.57 | 74.86 | 56.69 |
| Lucknow_Retail | 2 | 125.41 | 117.73 | 79.20 | 67.49 |
| | 3 | 194.25 | 143.68 | 84.73 | 65.61 |
| Lucknow_Wholesale | 2 | 390.74 | 164.18 | 92.79 | 74.23 |
| | 3 | 741.61 | 216.04 | 100.56 | 73.06 |
| Ludhiana_Retail | 2 | 262.12 | 937.45 | 179.97 | 84.53 |
| | 3 | 397.97 | 1233.07 | 214.37 | 88.56 |
| Ludhiana_Wholesale | 2 | 247.35 | 683.00 | 211.98 | 81.38 |
| | 3 | 360.43 | 901.09 | 260.63 | 85.69 |
| Maximum_Price_Retail | 2 | 493.20 | 175.26 | 96.65 | 71.00 |
| | 3 | 939.21 | 234.04 | 105.91 | 70.88 |
| Maximum_Price_Wholesale | 2 | 677.41 | 189.46 | 92.78 | 70.06 |
| | 3 | 1324.58 | 253.50 | 103.21 | 71.68 |
| Minimum_Price_Retail | 2 | 79.60 | 112.47 | 83.43 | 57.51 |
| | 3 | 141.35 | 146.67 | 92.67 | 57.06 |
| Minimum_Price_Wholesale | 2 | 80.12 | 76.01 | 60.34 | 44.10 |
| | 3 | 121.74 | 90.87 | 65.72 | 43.64 |
| Modal_Price_Retail | 2 | 118.78 | 118.55 | 66.71 | 47.41 |
| | 3 | 201.10 | 155.97 | 72.70 | 48.17 |
| Modal_Price_Wholesale | 2 | 143.71 | 60.58 | 42.94 | 36.59 |
| | 3 | 236.28 | 73.09 | 45.92 | 36.09 |
| Mumbai_Retail | 2 | 294.76 | 165.60 | 94.41 | 75.08 |
| | 3 | 547.07 | 217.46 | 102.39 | 74.63 |
| Mumbai_Wholesale | 2 | 129.13 | 132.31 | 89.78 | 66.80 |
| | 3 | 205.21 | 166.97 | 100.58 | 67.68 |
| Patna_Retail | 2 | 388.63 | 277.51 | 91.02 | 74.61 |
| | 3 | 733.61 | 396.26 | 98.42 | 74.56 |
| Patna_Wholesale | 2 | 199.47 | 191.24 | 84.96 | 61.60 |
| | 3 | 357.21 | 254.22 | 94.33 | 60.17 |
| Thiruvananthapuram_Retail | 2 | 203.09 | 111.99 | 81.55 | 64.48 |
| | 3 | 349.40 | 141.54 | 88.10 | 63.15 |
| Thiruvananthapuram_Wholesale | 2 | 224.43 | 227.39 | 102.92 | 67.89 |
| | 3 | 387.76 | 290.76 | 113.16 | 66.63 |

Note: All values of epsilons were statistically significant at the 1% level of significance.

**Table 6.** Asymmetric cointegration.

| Markets | MTAR | | | |
|---|---|---|---|---|
| | PHI | | APT | |
| | F Value | Pr. Value | F Value | Pr. Value |
| Ahmedabad | 14.60 | <0.05 | 3.55 | 0.05 |
| Amritsar | 6.03 | <0.05 | 2.43 | 0.11 |
| Bengaluru | 3.48 | <0.05 | 3.05 | 0.08 |
| Bhopal | 26.91 | <0.05 | 4.88 | <0.05 |
| Bhubaneswar | 6.38 | <0.05 | 0.02 | 0.88 |
| Chennai | 11.05 | <0.05 | 12.74 | <0.05 |
| Dehradun | 4.61 | <0.05 | 1.17 | 0.27 |
| Delhi | 13.33 | <0.05 | 4.44 | <0.05 |
| Hyderabad | 63.31 | <0.05 | 96.47 | <0.05 |
| Jaipur | 42.80 | <0.05 | 47.64 | <0.05 |
| Jammu | 49.47 | <0.05 | 38.20 | <0.05 |
| Lucknow | 40.52 | <0.05 | 32.62 | <0.05 |
| Ludhiana | 41.19 | <0.05 | 54.91 | <0.05 |
| Mumbai | 7.80 | <0.05 | 1.93 | 0.16 |
| Patna | 43.31 | <0.05 | 51.02 | <0.05 |
| Thiruvananthapuram | 17.38 | <0.05 | 1.13 | 0.28 |
| Maximum | 31.52 | <0.05 | 23.55 | <0.05 |
| Minimum | 35.72 | <0.05 | 41.16 | <0.05 |
| Modal | 20.84 | <0.05 | 4.82 | <0.05 |

PHI tests for the presence of cointegration while APT tests for the presence of asymmetric cointegration.

The acceptance of cointegration between two series implies that there exists a long-run relationship between them and means that an error correction model (ECM) is applicable, which combines the long-run relationship with the short-run dynamics of the model. Accordingly, the vector error correction model (VECM) was applied in order to find the speed of adjustment and long-run coefficient among wholesale and retail prices of wheat in individual markets. For fitting of the VECM model, first, an unrestricted VAR model was fitted and optimum lag was determined based on the Akaike information criterion (AIC) and Bayesian information criteria (BIC). The results of VECM is reported in Table S1 in the Supplementary Materials. In the markets of Delhi, Bhopal, Mumbai, Jaipur, Patna, and Bhubaneswar, the optimum number of lags for wholesale and retail prices was one lag; whereas in all the other markets, the optimum number of lags for wholesale and retail prices was found to be two. It was noted that most of the values of the error correction term (ECT) that are significant were found to be negative. The value of ECT signifies the speed of adjustment at which the market approaches equilibrium once it deviates. The speed of adjustment (per week) was found to be highest in Jaipur (18.8%), followed by Bhopal (17.4%), Lucknow (13%), and Hyderabad (13%) in retail price. However, in the case of wholesale price, the Jammu market had the highest speed of adjustment (13.2%) toward equilibrium.

For instance, the error correction model (ECM) for the Ahmedabad market can be written as

$$\Delta y_t^r = 0.059 - 0.096\, e_{t-1}^r + 0.305 \Delta y_{t-1}^r - 0.018 \Delta y_{t-2}^r + 0.120 \Delta y_{t-1}^w - 0.173 \Delta y_{t-2}^w$$

$$\Delta y_t^w = -0.034 + 0.057\, e_{t-1}^w + 0.284 \Delta y_{t-1}^w - 0.118 \Delta y_{t-2}^w + 0.041 \Delta y_{t-1}^r - 0.116 \Delta y_{t-2}^r$$

where $y_t^r$ and $y_t^w$ respectively denotes the retail and wholesale price at time t. $e_{t-1}^r$ and $e_{t-1}^w$ denotes the error correction term respectively for the retail and wholesale price equation.

Subsequent to applying VECM, the Sup-LM test [22] was applied to test the null hypothesis of linear cointegration against the two-regime threshold cointegration. The result of the Sup-LM is reported in Table 7. It can be seen from Table 7 that in the markets of Delhi, Ahmedabad, Bhopal, Hyderabad, Jammu, Mumbai, Patna, Thiruvananthapuram, all India minimum and modal price, the mechanism of the price transmission was not linear, but was rather of a threshold type.

**Table 7.** Result of the Sup-LM test.

| Markets | Test Statistic | *p* Value |
|---|---|---|
| Ahmedabad | 17.27 | 0.03 |
| Amritsar | 10.88 | 0.77 |
| Bengaluru | 11.30 | 0.49 |
| Bhopal | 21.54 | 0.01 |
| Bhubaneswar | 14.07 | 0.39 |
| Chennai | 10.70 | 0.56 |
| Dehradun | 13.72 | 0.35 |
| Delhi | 22.66 | 0.01 |
| Hyderabad | 18.90 | 0.02 |
| Jaipur | 15.69 | 0.29 |
| Jammu | 22.91 | 0.01 |
| Lucknow | 9.36 | 0.88 |
| Ludhiana | 11.55 | 0.75 |
| Mumbai | 17.97 | 0.03 |
| Patna | 19.00 | 0.02 |
| Thiruvananthapuram | 19.17 | 0.02 |
| Maximum | 11.29 | 0.81 |
| Minimum | 25.77 | 0.01 |
| Modal | 35.03 | 0.001 |

Accordingly, to accommodate the asymmetricity and nonlinearity in price transmission as clearly shown in Tables 5 and 6, threshold VECM was fitted with two regimes and the results are reported in Table S2 in the Supplementary Materials. The percentage of data points fell in the first regime and the second regime is mentioned in the second to last column in Table S2 and the threshold value as found in the individual market to divide the data in two regimes is mentioned in the last column of Table S2. The approach followed to find the optimum lag in the TVECM model was the same as that of the VECM model. The optimum value as obtained through the grid search is depicted in Figure 1. To save space, the optimum value of the threshold parameter and the cointegration parameter obtained through grid search only for the selected markets, namely, Ahmadebad, Amritsar, and Bhubaneswar are depicted in Figure 1. In the present investigation, in almost all markets, there was a significant difference in the percent of observations falling into the first and second regime except for Bhubaneswar and Chennai. In Bhubaneswar, 46.30% of observations fell in the first regime while 53.70% of observations fell in the second regime. Similarly, in Chennai, 45.50% of observations fell in the first regime and 54.50% of observations fell in the second regime. According to Hansen and Seo [22], it can be called a "typical" regime, where more than half of the observations belong to this regime, and the regime that includes a lower percent of the observations is known as an "extreme" regime. Therefore, the short-run dynamic effects of the retail and wholesale prices show significant differences between typical and extreme regimes.

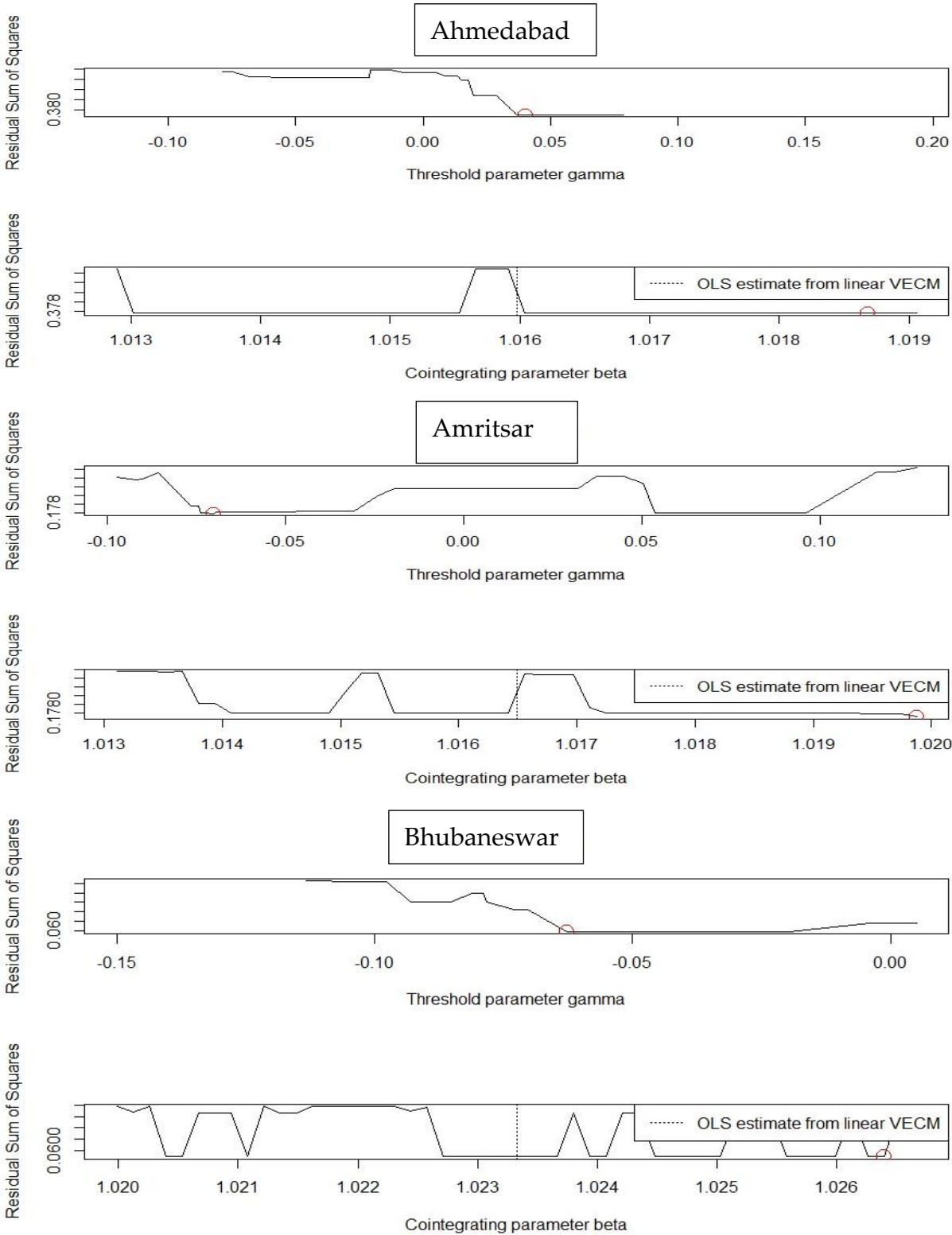

**Figure 1.** Grid search to find the optimum value of the threshold parameter and cointegration parameter in TVECM.

To this end, Granger causality [40] was computed among the wholesale and retail prices of different markets in India and the results are reported in Table S3. It was found that in most pairs of markets, the causality was bi-directional.

## 5. Discussion

The existence of both horizontal and vertical cointegration among the studied series implies that there exists a long-run relationship and an error correction model (ECM) that combines the long-run relationship with the short-run dynamics of the model. The results indicate that most of the error correction terms (ECTs) were statistically significant, implying that the system, once in disequilibrium, tries to come back to the equilibrium state (e.g., in Bhopal, the retail price once it deviates from equilibrium, approaches toward equilibrium at the rate of 9.6% per week (ECT = −0.096)). Furthermore, the existence of asymmetric cointegration was observed in more than 50% of the studied markets. Application of the two-regime TVECM demonstrated that the coefficient of ECT was significant in retail for both the regimes in Delhi; wholesale for both the regime and retail in typical regime for Jammu; retail and wholesale in the extreme regime for Amritsar; retail in both regimes, but wholesale in the extreme regime for Ludhiana; retail in the extreme regime in Lucknow; retail in both regimes for Dehradun; wholesale in a typical regime and retail in an extreme regime for Ahmedabad; retail in both regimes for Bhopal; retail in the extreme regime and wholesale in a typical regime for Mumbai; wholesale in a typical regime and retail in an extreme regime for Jaipur; retail and wholesale both in the extreme regime for Patna; retail and wholesale in both regimes; retail in the extreme regime for Bengaluru; retail in the typical regime for Thiruvananthapuram; wholesale in the second regime for Chennai; and retail in both regimes and wholesale in a typical regime for Hyderabad. This implies that retailers respond significantly to the deviations from the long-run equilibrium.

## 6. Conclusions

In the present study, presence of cointegration was tested by using Johansen's approach. It was revealed that wholesale and retail prices of wheat in all markets were cointegrated both horizontally as well as vertically. For horizontal integration, it was revealed that markets were spatially cointegrated with respect to wholesale as well as retail prices. This indicates that information on change in the wholesale/retail price in one market is transmitted to the other spatially integrated markets. In addition to the horizontal cointegration, the vertical cointegration between the wholesale and retail prices of wheat in individual markets was also investigated. It was observed that a change in wholesale price in a market is reflected in the change in retail price in the same market.

Asymmetricity in price transmission was investigated by means of TAR and M-TAR models. Application of the MTAR model revealed that most of the markets under consideration were asymmetric in terms of price transmission from wholesale to retail markets. Moreover, our findings pointed out that there are nonlinearities in the studied price adjustment process. To ensure asymmetricity as well as nonlinearity in cointegration and price transmission between the wholesale and retail prices of wheat, the TVECM model was applied. It can be seen that the price signals are transmitted across both the horizontal and vertical chain. However, the direction and intensity of price changes may be affected by the dynamic linkages between the demand and supply. The results from the study will help improve the precision of the information to predict the price movements used by marketing operators to formulate appropriate strategies. As the forecast of the wholesale (retail) price of a commodity will not only depend on changes in its own lag prices, but also on the changes in lags of the retail (wholesale) prices of that commodity. Similarly, for prediction of the wholesale price of a commodity in a market, the lagged wholesale price of that market along with lagged prices of other integrated markets need to be considered.

The study will help policy makers design suitable marketing strategies in bringing efficiency in agricultural markets. It is noted that price changes are temporary and would converge to an equilibrium within a given time span. Consideration of proper domestic supply management and international trade, along with strong market surveillance, will minimize the gap between the wholesale and retail prices of agricultural commodities. The present study has some limitations as this did not highlight the factors that affect the price of the commodity in a cointegraton study. This may be explored in future research.

Furthermore, in studying vertical cointegration, the farm harvest price of the commodity may be included.

**Supplementary Materials:** The following supporting information can be downloaded at: https://www.mdpi.com/article/10.3390/agriculture12030410/s1, Table S1: Results of VECM model; Table S2: Results of TVECM model; Table S3: Granger Causality test results.

**Author Contributions:** Conceptualization, R.K.P. and T.K.; Formal analysis, R.K.P.; Investigation, R.K.P.; Methodology, R.K.P. and T.K.; Writing—review and editing, R.K.P. and T.K. All authors have read and agreed to the published version of the manuscript.

**Funding:** This research received no external funding.

**Informed Consent Statement:** Not applicable.

**Data Availability Statement:** The data were collected from the Department of Consumer Affairs, Government of India.

**Acknowledgments:** The authors are grateful to the two anonymous reviewers for their extensive comments that helped in improving the quality of this manuscript.

**Conflicts of Interest:** The authors declare that there are no conflict of interest in publishing this manuscript.

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
