# Peer review of "Asymmetric Price Transmission: A Case of Wheat in India"

_agriculture, doi:10.3390/agriculture12030410_

Round 1
Reviewer 1 Report
Thank you for letting me read this interesting article.
First of all, I do not really understand why the authors decided to cite old references to cointegration. I also see that they use some pioneering research, but the newer article still lacks some more advanced and recent citations (e.g. 10-15 of them), e.g., https://doi.org/10.5513/jcea.v13i1.1073 and https://doi.org/10.1002/jtr.2163 or https://doi.org/10.1080/1331677X.2021.2019078, https://doi.org/10.1108/BFJ-03-2014-0109. In addition, there are too many self-citations (No. 48, 47, 35, 34, 33). Also, I would like to see the motivation for the present study and the aim(s), objective(s), and goal(s) of the study stated in the introduction, along with at least one hypothesis. Also, some spelling corrections are needed in the introduction, for example in line 81 between the period and although.
Second, I can not see in the methodology section that impulse response analysis is useful and therefore could be dropped from the study. It seems to have been included as a learning parameter for students, not for the scientific purpose of the article (there are no hypotheses or goals associated with this method). In line 174, perhaps r=k-1 could be inserted?
In line 163, replace statistical techniques with econometric methods. Also, please explain ALL the symbols used in the Data and Methodology section. It is not clear to me how you are going to check or use lag length and how you are going to check normalities in the series. These are two important issues that I would like to see explained in the methodology section.
Third, is it possible to determine the "language" you are using in Table 1? Nominal, real or relative prices, in what currency? Why do we need Figure 1 if we have summary statistics and no weighted coefficients? Figure 1 can be omitted. Perhaps some long tables (e.g. Table 9) should be moved to the appendices and only a brief overview could be included in the main text, or as an equation (as with Table 7, write VECM and leave the table in the appendix). Line 308 is simply JUMP in the text. Is there a better way to insert it? I have not found the table S1, L. 346!!!
Fourth, instead of the impulse responses in Fig. 3, insert a discussion section, which is now missing.
And finally, the conclusions are condensed into a single paragraph and are difficult to read and understand. Please revise. Can you include vertical and horizontal integration in a separate section in the conclusions and include a section with the theoretical and empirical contribution? At the moment, the latest (empirical) findings are mainly described in the conclusions.
Author Response
Reply to reviewer’s comments:
We sincerely thank the Editor and the reviewers for their extensive comments. These comments have been extremely useful in reshaping the paper by identifying the weak points in the paper and providing roadmap for reorganizing the paper. Below we give replies to the comments as to how we have tried to address the issues raised by the reviewers. We have attempted to handle all the comments and revised the document to meet the reviewers’ requirements that we think have improved the paper significantly in conjunction.
Please note that the reviewers’ comments are in red, our responses are in blue.
Reviewers' comments:
Reviewer #1:
Comment: First of all, I do not really understand why the authors decided to cite old references to cointegration. I also see that they use some pioneering research, but the newer article still lacks some more advanced and recent citations (e.g. 10-15 of them), e.g., https://doi.org/10.5513/jcea.v13i1.1073 and https://doi.org/10.1002/jtr.2163 or https://doi.org/10.1080/1331677X.2021.2019078, https://doi.org/10.1108/BFJ-03-2014-0109.
In addition, there are too many self-citations (No. 48, 47, 35, 34, 33). Also, I would like to see the motivation for the present study and the aim(s), objective(s), and goal(s) of the study stated in the introduction, along with at least one hypothesis. Also, some spelling corrections are needed in the introduction, for example in line 81 between the period and although.
Authors’ reply: As suggested by the reviewer, the newer articles have been included in the revised manuscript. The old references about cointegration have been removed. Also, the self-citations have been reduced. The aim, objective, and goal of the study has been incorporated in the background section (Section 2) of the revised manuscript. All the spellings have corrected.
Comment: Second, I can not see in the methodology section that impulse response analysis is useful and, therefore, could be dropped from the study. It seems to have been included as a learning parameter for students, not for the scientific purpose of the article (there are no hypotheses or goals associated with this method). In line 174, perhaps r=k-1 could be inserted?
In line 163, replace statistical techniques with econometric methods. Also, please explain ALL the symbols used in the Data and Methodology section. It is not clear to me how you are going to check or use lag length and how you are going to check normalities in the series. These are two important issues that I would like to see explained in the methodology section.
Authors’ reply: As suggested by the reviewer, the impulse response analysis has been dropped from the study. All the symbols used in the Data and Methodology section have been explained. The lag length selection is based on minimum AIC and BIC, and the same is mentioned in the revised manuscript. The Jarque-Bera test checks the normality of the series, and it is reported in the revised manuscript.
Comment: Third, is it possible to determine the "language" you are using in Table 1? Nominal, real or relative prices, in what currency? Why do we need Figure 1 if we have summary statistics and no weighted coefficients? Figure 1 can be omitted. Perhaps some long tables (e.g. Table 9) should be moved to the appendices and only a brief overview could be included in the main text, or as an equation (as with Table 7, write VECM and leave the table in the appendix). Line 308 is simply JUMP in the text. Is there a better way to insert it? I have not found the table S1, L. 346!!!
Authors’ reply: The accurate prices used in the present study and the same are mentioned in table 1. The currency is also mentioned. As suggested, figure 1 has been removed. Tables 7 and 9 have been moved to appendices, and a brief overview is included in the main text. The text is modified, and table S1 is given in the appendix.
Comment: Fourth, instead of the impulse responses in Fig. 3, insert a discussion section, which is now missing.
Authors’ reply: Figure 3 has been removed, and a discussion section is inserted in the revised manuscript.
Comment: And finally, the conclusions are condensed into a single paragraph and are difficult to read and understand. Please revise. Can you include vertical and horizontal integration in a separate section in the conclusions and include a section with the theoretical and empirical contribution? At the moment, the latest (empirical) findings are mainly described in the conclusions
Authors’ reply: Conclusion has been revised. A separate paragraph is made towards vertical and horizontal integration and separate paragraph is written on theoretical and empirical contribution.

Reviewer 2 Report
Asymmetric Price Transmission: A Case of Wheat in India
This paper makes a horizontal and vertical analysis of integration in wholesale and retail price of wheat in major markets of India.
This study is relevant due the importance of wheat market in India and the world. Changes in this market can affect to international balance.
In summary, the economic interpretation of results should be improved.
Introduction and Background:
In Introduction you say “Major causes of asymmetric price transmission are: the presence of non-competitive markets and existence of adjustment costs (Meyer and Von 80 Cramon‐Taubadel, 2004)”. Please review also Pérez- Mesa et al. (2010) “Retail price rigidity in perishable food products: a case study”.
Data and Methodology:
The analysis carry out is standard (Johansen’s approach of cointegration; Asymmetric cointegration). Is there any novelty?
Results and Discussion:
Please, yo have to reduce the size of Figure 1. Perhaps it should be located in Methodology.
In table 2, Is the difference in wholesale and retail prices well explained by the costs and benefits applied in the chain?
Columns of p-value in Table 2 can be removed and referenced to p-value in the bottom of the figure.
You have to find a way to reduce the number of tables and show only the most relevant aspects. Even inside the tables reduce its size.
About Table 7, you say “The value of ECT signifies the speed of adjustment at which the market approaches to the equilibrium once it deviates.” Please, you have interpret the results economically (not just statistically). There is no real economic analysis (interpretation) of what the statistical results mean. The same in Table 8.
In short, there is no discussion.
Conclusions
There is no political or managerial implication. The asymmetry exists, so does the long-term equilibrium. It does not explain what implications this may have for producers. It does not explain what implications the results may have for the market as a whole. It is merely a statistical exercise.
All the best
Author Response
Reply to reviewer’s comments:
We sincerely thank the Editor and the reviewers for their extensive comments. These comments have been extremely useful in reshaping the paper by identifying the weak points in the paper and providing roadmap for reorganizing the paper. Below we give replies to the comments as to how we have tried to address the issues raised by the reviewers. We have attempted to handle all the comments and revised the document to meet the reviewers’ requirements that we think have improved the paper significantly in conjunction.
Please note that the reviewers’ comments are in red, our responses are in blue.
Reviwer-2
Introduction and Background:
Comment: In Introduction you say “Major causes of asymmetric price transmission are: the presence of non-competitive markets and existence of adjustment costs (Meyer and Von 80 Cramon‐Taubadel, 2004)”. Please review also Pérez- Mesa et al. (2010) “Retail price rigidity in perishable food products: a case study”.
Authors’ reply: The paper “Retail price rigidity in perishable food products: a case study” by Pérez- Mesa et al. (2010) has been reviewed, and the same is cited in the revised manuscript
Data and Methodology:
Comment: The analysis carry out is standard (Johansen’s approach of cointegration; Asymmetric cointegration). Is there any novelty?
Authors’ reply: The methodology is standard, but there is very little application of Asymmetric cointegration in agriculture in general and in Indian agriculture in particular. Therefore, the study was taken to investigate the asymmetric cointegration in one of the most staple food of India i.e., wheat, among the major markets.
Results and Discussion:
Comment: Please,youo have to reduce the size of Figure 1. Perhaps it should be located in Methodology.
Authors’ reply: Figure 1 has been removed from the revised manuscript.
Comment: In table 2, Is the difference in wholesale and retail prices well explained by the costs and benefits applied in the chain?
Authors’ reply: The difference in wholesale and retail price is explained in the text.
Comment: Columns of p-value in Table 2 can be removed and referenced to p-value in the bottom of the figure.
Authors’ reply: Columns of p-value in Table 2 has been removed and significant values are marked with *
Comment: You have to find a way to reduce the number of tables and show only the most relevant aspects. Even inside the tables reduce its size.
Authors’ reply: The number of tables have been reduced
Comment: About Table 7, you say, “The value of ECT signifies the speed of adjustment at which the market approaches to the equilibrium once it deviates.” Please, you have interpret the results economically (not just statistically). There is no f economic analysis (interpretation) of what the statistical results mean. The same in Table 8.
Authors’ reply: Tables 7, 9 have been moved to an appendix. The result of these tables has been discussed in the text.
Comment: In short, there is no discussion.
Authors’ reply: One sub-heading discussion has been inserted in the revised manuscript
Conclusions
Comment: There is no political or managerial implication. The asymmetry exists, so does the long-term equilibrium. It does not explain what implications this may have for producers. It does not explain what implications the results may have for the market as a whole. It is merely a statistical exercise.
Authors’ reply: the conclusion is revised, considering the above-suggested points.

Round 2
Reviewer 1 Report
Thank you.
I still can not see Figure S1 in the attachment.
I think there is still room for 6 citations (references).
Otherwise, the research has been improved.
Good luck!
Reviewer
Author Response
Reply to reviewer’s comments:
We sincerely thank the Editor and the reviewers for their extensive comments. These comments have been extremely useful in reshaping the paper by identifying the weak points in the paper and providing roadmap for reorganizing the paper. Below we give replies to the comments as to how we have tried to address the issues raised by the reviewers. We have attempted to handle all the comments and revised the document to meet the reviewers’ requirements that we think have improved the paper significantly in conjunction.
Please note that the reviewers’ comments are in red, our responses are in blue.
Reviewers' comments:
Reviewer #1:
Comment: I still can not see Figure S1 in the attachment.
Authors’ reply: It is table S1, which is the result of VECM model and is given in appendix (supplementary materials)
Comment: I think there is still room for 6 citations (references).
Authors’ reply: The following references have been added in the revised manuscript
- Aghabeygi, M.; Antonioli, F.; Arfini, F. Assessing symmetric price transmission by using threshold cointegration in Iranian egg market. British Food Journal. 2021, 123 (6), 2278-2288.
- Rezitis, A. N.; Rokopanos, Andreas. Asymmetric Price Transmission along the European Food Supply Chain and the CAP Health Check: a Panel Vector Error Correction Approach. Journal of Agricultural & Food Industrial Organization. 2019, 17 (2), 20180002.
- Deb, L; Lee, Y.; Lee, S.H. Market Integration and Price Transmission in the Vertical Supply Chain of Rice: An Evidence from Bangladesh. Agriculture. 2020, 10, 271.
- Wiseman,T.; Luckstead, J.; Durand-Morat, A. Asymmetric Exchange Rate Pass-Through in Southeast Asian Rice Trade, Journal of Agricultural and Applied Economics. 2021, 53, 341–374.
- Mallick, L.; Behera, S.R.; Murthy, R.V.R. Dynamics of capital account and current account in India: Evidence from threshold cointegration with asymmetric error correction, Applied Economics Letters. 2021, DOI: 1080/13504851.2021.1990836.
- Bagnai, A.; Opsina, C.A.M. Asymmetries, Outliers, and Structural Stability in the US Gasoline Market. Energy Economics. 2018, 69, 250–60.

Reviewer 2 Report
I don't agree in answer authors "There is very little application of Asymmetric cointegration in agriculture Asymmetric cointegration in agriculture" Maybe yes in Indian agriculture. Please, review the literature in this regard
In conclusions you say “The results from the study will help improve the information precision to predict the price movements used by marketing operators for formulating appropriate strategies. The study will help policy makers in order to design suitable marketing strategies in bringing efficiency in agricultural markets”. In this sense, please you must present at least one practical example.
Author Response
Reply to reviewer’s comments:
We sincerely thank the Editor and the reviewers for their extensive comments. These comments have been extremely useful in reshaping the paper by identifying the weak points in the paper and providing roadmap for reorganizing the paper. Below we give replies to the comments as to how we have tried to address the issues raised by the reviewers. We have attempted to handle all the comments and revised the document to meet the reviewers’ requirements that we think have improved the paper significantly in conjunction.
Please note that the reviewers’ comments are in red, our responses are in blue.
Reviewers' comments:
Reviewer #2:
Comment: I don't agree in answer authors "There is very little application of Asymmetric cointegration in agriculture " Maybe yes in Indian agriculture. Please, review the literature in this regard.
Authors’ reply: Thanks for the suggestion. It is in Indian agriculture where we could not find much study on asymmetric cointegration. The literature has been reviewed and following references have been added in the revised manuscript.
- Aghabeygi, M.; Antonioli, F.; Arfini, F. Assessing symmetric price transmission by using threshold cointegration in Iranian egg market. British Food Journal. 2021, 123 (6), 2278-2288.
- Rezitis, A. N.; Rokopanos, Andreas. Asymmetric Price Transmission along the European Food Supply Chain and the CAP Health Check: a Panel Vector Error Correction Approach. Journal of Agricultural & Food Industrial Organization. 2019, 17 (2), 20180002.
- Deb, L; Lee, Y.; Lee, S.H. Market Integration and Price Transmission in the Vertical Supply Chain of Rice: An Evidence from Bangladesh. Agriculture. 2020, 10, 271.
- Wiseman,T.; Luckstead, J.; Durand-Morat, A. Asymmetric Exchange Rate Pass-Through in Southeast Asian Rice Trade, Journal of Agricultural and Applied Economics. 2021, 53, 341–374.
- Mallick, L.; Behera, S.R.; Murthy, R.V.R. Dynamics of capital account and current account in India: Evidence from threshold cointegration with asymmetric error correction, Applied Economics Letters. 2021, DOI: 1080/13504851.2021.1990836.
- Bagnai, A.; Opsina, C.A.M. Asymmetries, Outliers, and Structural Stability in the US Gasoline Market. Energy Economics. 2018, 69, 250–60.
Comment: In conclusions you say “The results from the study will help improve the information precision to predict the price movements used by marketing operators for formulating appropriate strategies. The study will help policy makers in order to design suitable marketing strategies in bringing efficiency in agricultural markets”. In this sense, please you must present at least one practical example.
Authors’ reply: The forecast of wholesale (retail) price of a commodity will not only depend on changes on its own lag prices but also on the changes in lags of retail (wholesale) prices of that commodity. Similarly, for prediction of the wholesale price of a commodity in a market, the lagged wholesale price of that market along with lagged prices of other integrated markets need to be considered. Also, the dominating market for a particular commodity based on the Granger causality can be determined which governs the price of that commodity in most of the markets and accordingly, the policy makers may find proper strategy to stabilize the information movement.
